# Snow Albedo Seasonality and Trend from MODIS Sensor and Ground Data at Johnsons Glacier, Livingston Island, Maritime Antarctica

**DOI:** 10.3390/s19163569

**Published:** 2019-08-15

**Authors:** Javier F. Calleja, Alejandro Corbea-Pérez, Susana Fernández, Carmen Recondo, Juanjo Peón, Miguel Ángel de Pablo

**Affiliations:** 1Remote Sensing Applications (RSApps) Research Group, Department of Physics, University of Oviedo, 33007 Oviedo, Spain; 2Remote Sensing Applications (RSApps) Research Group, Department of Mining Exploitation and Prospecting, University of Oviedo, 33600 Mieres, Spain; 3Mathematical Modelling (MOMA) Research Group, Department of Geology, University of Oviedo, 33005 Oviedo, Spain; 4Department of Geology, Geography and Environment, University of Alcalá, 28805 Alcalá de Henares, Spain

**Keywords:** albedo, Antarctica, MODIS snow albedo, in-situ albedo, albedo trend, albedo seasonality

## Abstract

The aim of this work is to investigate whether snow albedo seasonality and trend under all sky conditions at Johnsons Glacier (Livingston Island, Antarctica) can be tracked using the Moderate Resolution Imaging Spectroradiometer (MODIS) snow albedo daily product MOD10A1. The time span is from December 2006 to February 2015. As the MOD10A1 snow albedo product has never been used in Antarctica before, we also assess the performance for the MOD10A1 cloud mask. The motivation for this work is the need for a description of snow albedo under all sky conditions (including overcast days) using satellite data with mid-spatial resolution. In-situ albedo was filtered with a 5-day windowed moving average, while the MOD10A1 data were filtered using a maximum filter. Both in-situ and MOD10A1 data follow an exponential decay during the melting season, with a maximum decay of 0.049/0.094 day^−1^ (in-situ/MOD10A1) for the 2006–2007 season and a minimum of 0.016/0.016 day^−1^ for the 2009–2010 season. The duration of the decay varies from 85 days (2007–2008) to 167 days (2013–2014). Regarding the albedo trend, both data sets exhibit a slight increase of albedo, which may be explained by an increase of snowfall along with a decrease of snowmelt in the study area. Annual albedo increases of 0.2% and 0.7% are obtained for in-situ and MOD10A1 data, respectively, which amount to respective increases of 2% and 6% in the period 2006–2015. We conclude that MOD10A1 can be used to characterize snow albedo seasonality and trend on Livingston Island when filtered with a maximum filter.

## 1. Introduction

Albedo is defined as the bi-hemispherical reflectance, which is the ratio of the radiant flux reflected from a unit surface area into the whole hemisphere to the incident radiant flux of hemispherical angular extent [1]. Intensive research has been carried out to obtain the albedo time evolution in the Arctic and Antarctic regions, areas which are highly sensitive to climate change [2,3,4,5,6]. Albedo maps of the entire polar areas have been obtained using remote sensing data and, in the case of Antarctica, the highest spatial resolution of the albedo maps obtained up to date is 5 km [5,7]. Although this spatial resolution is appropriate for climate studies on a regional or global scale, some studies demand a knowledge of albedo and albedo time evolution at a higher spatial resolution. Most of Antarctica is permanently covered by ice and snow and ice melting is known to occur only in areas close to the coastline during summer [7,8]. These areas are the most interesting ones for some studies, which require a description of the snow cover with a spatial resolution higher than 5 km. Some examples of such studies are: The impact of snow cover duration and summer snowmelt on lichen populations on Livingston island [9], the monitoring of the thickness of the permafrost active layer [10,11,12,13,14,15], or the study of the surface energy and mass balance of glaciers [16,17].

If the incoming radiation is divided into a direct and a diffuse component, and the diffuse part is assumed to be isotropic, the albedo *α* can be calculated as:(1)$$\alpha =d\times \alpha_{dir}+\left(1-d\right)\times \alpha_{diff},$$
where *α**_dir_* is the directional-hemispherical reflectance, *α**_diff_* is the bi-hemispherical reflectance for pure diffuse isotropic irradiance, and *d* is the fractional amount of direct irradiance. When comparing albedo from orbital sensors against in-situ data, we must take into account the nature of each data set. In-situ albedo, measured with two pyranometers (one facing the sky and another facing the Earth’s surface, both of them parallel to the surface), corresponds to bi-hemispherical reflectance (α in Equation (1)). The product provided by orbital sensors can be the directional-hemispherical reflectance (*α**_dir_* in Equation (1)), also called the black-sky albedo, or the pure bi-hemispherical reflectance (*α**_diff_* in Equation (1)), also called the white-sky albedo. For instance, the albedo product developed for the Advanced Very High Resolution Radiometer (AVHRR) is directional-hemispherical reflectance [18]. In the case of the Moderate Resolution Imaging Spectroradiometer (MODIS), the so-called albedo provided by the daily snow albedo product MOD10A1 is also the directional-hemispherical reflectance [19,20], while the 16-day albedo product MCD43 provides *α**_dir_* and *α**_diff_* separately [21]. Once *α**_dir_* and *α**_diff_* are known, *d* can be estimated using the Aerosol Optical Depth product MOD04 of MODIS [22], or by using in-situ measurements from a close station [23]. When comparing albedo from orbital sensors against in-situ data, only in-situ data under clear-sky conditions are taken into account, while overcast days are disregarded [19]. This makes sense, since optical satellite data of the Earth’s surface can only be captured when the sky is clear. It is well-known that cloud cover normally increases the spectrally integrated albedo of snow [24,25], so a description of albedo evolution over time considering clear-sky days only will very probably provide biased results. In addition to this, restricting the analysis to clear-sky days will dramatically diminish the amount of available data.

In this scenario, the aim of this work is to compare the snow albedo seasonality and the snow albedo trend in the period of 2006–2015 using MODIS data and in-situ data. The Moderate Resolution Imaging Spectroradiometer (MODIS) on board the Terra and Aqua platforms has two daily snow albedo products (MOD10A1 from the Terra sensor and MYD10A1 from the Aqua sensor) [26] and a daily albedo product and MCD43 obtained combining information from Terra and Aqua [27]. All of these products have a spatial resolution of 500 m, but none of them have been used on Antarctica before. These daily albedo products have been used intensively in the northern hemisphere [21,28,29,30]. For this investigation, we chose the MOD10A1 (V005) snow daily product, for reasons explained below. The question we intend to answer in this work is: Can the seasonality and the trend of in-situ snow albedo (including all sky conditions) be tracked using the directional-hemispherical reflectance data from MODIS? We will prove that, if the MODIS data are filtered using a maximum filter, then the answer is yes.

This paper is organized as follows: In Section 2, we describe the materials and methods. Materials include the description of the study area, the in-situ data, and the satellite data, with special emphasis on explaining the reasons for the adoption of the MODIS MOD10A1 (V005) product. Methods include the description of the data processing separately for the cloud mask, the albedo filtering, the albedo seasonality, and the albedo trend. In Section 3, the results and discussion are presented separately for the cloud mask performance, the diurnal evolution of in-situ albedo, the albedo seasonality, and the albedo trend. Conclusions are summarized at the end of the paper.

## 2. Materials and Methods

### 2.1. Study Area

The study area is a site on Livingston Island, in the South Shetland Islands (SSI) Archipelago in Antarctica, which includes King George, Nelson, Robert, Greenwich, Livingston, Deception, Snow, Low, and Smith islands (Figure 1). The SSI Archipelago, with an area of 3300 km^2^, is separated from the Antarctic Peninsula by the Bransfield Strait and from South America by the Drake Passage. Livingston Island, the second largest island (974 km^2^) of the archipelago, is 110 km to the northwest of Cape Roquemaurel, mainland Antarctica. Most of the island’s surface (90%) is permanently covered with ice and only the west region and some coastal areas are snow-free during summer.

### 2.2. In-Situ Data

Two automatic weather stations (AWS) operated by the Spanish Meteorological Agency (Agencia Estatal de Meteorología, AEMET) are located on Livingston Island: Juan Carlos I (JCI) and Johnsons Glacier (JG). The stations are 1500 m apart; Juan Carlos I is 50 m from the coastline and Johnsons Glacier is inland. The JCI AWS is located at 62° 39′48″ S and 60° 23′19″ W at South Bay, 13 m above sea level [31]. At JG, an automatic AWS Campbell CR3000 was installed in December 2006. It was located at 62° 40′16″ S and 60° 21′ 51″ W, 178 m above sea level. In February 2015 this AWS was transferred to the Hurd Glacier. Hence, data from JG are available from December 1 2006 to February 11 2015. The JCI data indicate an average annual temperature of –1.2 °C between 1988 and 2014, with maximum value of 15.5 °C in summer and minimum temperature of –22.6 °C in winter. The mean annual relative humidity is 83%, the average atmospheric pressure is relatively low (988.7 hPa), and the average number of precipitation days is 162 per year; precipitation is mostly solid, in the form of snow or granular snow, although, in the summer, liquid precipitation may be frequent.

JG provides albedo data, but not diffuse, direct. and global irradiances separately. Albedo was measured at JG using two Kipp–Zonen CNR-1 pyranometers, one for incident radiation and another for reflected radiation. Albedo data are provided every 10 min. The pyranometers are placed 3 m above the snow surface. It is worth mentioning that, in these instruments, the diameter of the circular area that contributes 99% of the measured flux is approximately 10 times the sensor height [32], which is 30 m in this case. Albedo exhibits a great dependence on the Sun Zenith Angle (SZA) [25], and measurements with SZA > 75° are not reliable [33]. Only dates with a sufficient number of data with SZA < 75° are considered in the analysis, considering as representative those days with SZA < 75° two hours before and after local solar noon (from 10 to 14 h local solar time (LST). For the study area, this occurs from September 1 to April 10. The mean daily albedo is then calculated as the mean value from 10 to 14 h LST, minimizing the snow albedo variations due to variations of SZA [16,25]. The total number of days with in-situ albedo data at JG was 1008 (see Table 1). The range of dates with in-situ albedo data at JG each season is given in Table 2.

JCI provides albedo, diffuse, direct, and global radiation every 30 min. This station is equipped with Kipp–Zonen CM11 sensors (for global and diffuse radiation) and Kipp–Zonen CH1 (for direct radiation). Diffuse and global radiation at JCI are only available for summer days in the range of dates shown in Table 3. The range of dates is included within the interval between September 1 and April 10 for all seasons. The total number of days with diffuse and global irradiance data at JCI was 557.

The pyranometers of both stations are replaced every two years with calibrated ones.

### 2.3. Satellite Data

There are three MODIS products providing snow albedo on a daily basis with a spatial resolution of 500 m: MOD10A1, MYD10A1, and MCD43. MOD10A1 and MYD10A1 are daily snow albedo products obtained from the MODIS sensor onboard the Terra and Aqua satellites, respectively [26]. In the first steps of the algorithm used to calculate the albedo, MODIS band 6 (centered at 1600 nm) is used. This band is not fully functional on Aqua [26], and this is the main reason why the use of MYD10A1 was ruled out at the beginning of this work. The MCD43 BRDF/NBAR/Albedo (V006) data is retrieved on a daily basis [34]. This product provides both direct hemispherical reflectance and bi-hemispherical reflectance for MODIS bands 1–7. An inversion of the bi-directional reflectance distribution function (BRDF) is performed using all available observations during the 16-day moving window centered on the date of interest (9th day of the 16-day interval). If there are no sufficient cloud-free, high-quality observations for the BRDF inversion, a backup algorithm is used; in such a case, the datum is flagged as low-quality. The number of days with high-quality spectral albedo for bands 1–7 from MCD43 (V006) for the pixel where JG was located in the time span December 2006–February 2015 was 0. The study area is very cloudy, making it very difficult to obtain a sufficient number of clear days in a 16-day span. In addition to this, some authors [35,36] have found that the Ross Thick-Li Sparse Reciprocal (RTLSR) model is not necessarily appropriate for modeling snow BRDFs and, moreover, presented that the RTLSR model somewhat underestimates snow albedo, to some degree. Therefore, we decided to use MOD10A1. The latest version of the MOD10A1 product is Version 6 (V006). In this work, we used Version 5 (V005) because MOD10A1 had never been used in Antarctica before and the scientific literature was more abundant. Upgrading from V5 to V6 has been shown to correct a drift in snow albedo trend over the Greenland Ice Sheet [37]. However, in that work, the snow albedo trend of V005 was compared against the snow albedo trend of V006, with no in-situ data. In the present work, the results obtained on Livingston Island using V005 were compared against in-situ data. In addition to this, the results were similar to those obtained using V006 (preliminary results are provided as Appendix A). It is worth noting that MODIS albedo products have been used with great caution in the southern hemisphere. The reason for this seems to be the poor accuracy of the cloud mask over snow-covered areas, as pointed out by Bormann et al. [38] in Australia and Sirguey et al. [39] in New Zealand. In the northern hemisphere, the main source of cloud/snow confusion has been attributed to snow pack edges [40] while, in Australia, the greatest cloud/snow confusion rates are found within snow-covered areas [38]. These results indicate that the ultimate reason for the poor performance of the cloud mask in the southern hemisphere is unclear. While some authors have addressed the problem by devising a new cloud mask algorithm [41], in this work we intend to carry out an assessment of the cloud mask output directly. As this product is going to be used on an Antarctic site for the first time, it is important to have data from previous investigations for comparison. Otherwise, we would not be able to know if the differences observed were due to the study area or to the MOD10A1 version being used. For all of these reasons, we used MOD10A1 (V005).

The MODIS MOD10A1(V5) product data was downloaded using the Google Earth Engine platform [42]. The data used in this work corresponded to the pixel at which JG was placed. Unfortunately, the AWS at JCI is located close to the sea, and the MODIS pixel where this AWS was located included sea water, which produced errors in the classification of the pixel, being sometimes classified as ocean. In order to avoid this problem, in-situ albedo data will be compared with MOD10A1 data only at JG location. The daily snow albedo product MOD10A1 consists of four layers: Daily type of cover (snow, cloud, no snow, night, or others), daily snow cover fraction, daily snow albedo, and a quality assessment flag. The snow albedo layer provides the value of the snow albedo (0–100) when the pixel is classified as snow, or other classifications, which are: 101 = no decision, 111 = night, 125 = land, 137 = inland water, 139 = ocean, 150 = cloud, 250 = missing, 251 = self-shadowing, 252 = landmask mismatch, 253 = BRDF_failure, and 254 = non-production mask. Several filters were applied to the data. First of all, from the snow albedo layer, only those days classified as Land, Snow, or Cloud were selected. Secondly, the MOD10A1 quality filter was applied, keeping only those days for which the Snow_Spatial_QA Field was zero (high quality). Finally, the SZA filter was applied, only taking into account those days with SZA < 75° two hours before and after local noon (from September 1 to April 10). Out of the total number of days between December 1 2006 and February 11 2015, after the application of all the filters, the number of days with MOD10A1 data classified as Snow, Land, Cloud, was 1546 (Table 1), which were the days used in the assessment of the cloud mask. The total number of days with both MOD10A1 data (Snow, Land, Cloud) and in-situ cloud index data (clr) was 464 (See Section 2.4.1). Out of the 1546 days classified as Snow, Land, or Cloud, snow albedo was only given for those classified as Snow, which was 286 days. Finally, the number of days with both in-situ and MOD10A1 albedo data was 159, as shown in Table 1. The range of dates with MOD10A1 albedo data at JG for each season is shown in Table 2.

In order to complete the assessment of the MOD10A1 cloud mask, we also obtained 22 Landsat 7 and 62 Landsat 8 images, whose acquisition dates coincided with those for which MODIS data classified as Snow, Land, or Cloud were available. A few representative cases were also checked by visual inspection, using MOD09GA images.

### 2.4. Data Processing

#### 2.4.1. Cloud Mask

From JCI, only global and diffuse irradiance data were used in order to calculate the in-situ cloud index (clr), as explained below. We used the cloud index calculated with the data from JCI to characterize the cloud cover at JG. Radiation and meteorological data from JCI has been considered to be representative of other locations on Livingston Island, including JG [16,31]. Diffuse (dif) and direct (dir) radiation data have been used to study daily and annual albedo variations in Antarctica and the Arctic [33]. In the present study, the mean daily cloud index (clr) was calculated using daily mean values of diffuse and global radiation, using the equation:(2)$$clr=\frac{dif}{global},$$

If clr > 0.7, the day was classified as Cloud; otherwise, as Clear. For a completely overcast day, the diffuse radiation would be maximal and the direct radiation would be minimal, such that clr = 1.

The MOD10A1 cloud mask output was compared to the standardized method based on the Normalized Difference Snow Index (NDSI), using Landsat 7 and Landsat 8 images. In the case of Landsat data, two threshold values for the NDSI were considered: 0.4 and 0.7. Landsat images were, first, converted to Top of Atmosphere (TOA) reflectance. Subsequently, the DOS (Dark Object Subtraction) algorithm [43] was applied to homogenize the TOA reflectance image, as previously done in snow-covered areas with World View-2 data in Antarctica [44] and Landsat 7 images over Patagonia [45]. Dark pixels were selected from the Drake Passage, where depths above 4000 m are reached in several areas. Then, the NDSI was calculated [46,47] using the equation:(3)$$NDSI=\frac{R_{1}-R_{2}}{R_{1}+R_{2}},$$
where *R*_1_ is the DOS-corrected band 2 (485 nm) for L7 and band 3 (561 nm) for L8, and *R*_2_ is the DOS-corrected band 5 (1650 nm) for L7 and band 6 (1608 nm) for L8. This index is based on the fact that snow is highly reflective in the visible spectrum, but it reflects very little in the Shortwave Infrared (SWIR) range. For Landsat 8 data, the mean value of the NDSI was calculated on a 17 × 17 window, in order to obtain the same pixel size as MODIS (500 m). It was not possible to calculate the mean value over a 17 × 17 pixel window in the case of Landsat 7, due to the presence of data gaps caused by the Scan Line Corrector failure [48]; this is why, in the case of Landsat 7, we used the value of the NDSI on the pixel where JG is located, which, fortunately, lay on a pixel with data. NDSI threshold values have been used to discriminate cloud and snow. A threshold of 0.4, above which the pixel is classified as snow, has been proposed by Hall et al. [47]. Other authors [49] have taken 0.6 as a threshold, based on the fact that the optimal threshold of the snow cover varies depending on the season of the year in a range between 0.4 and 0.6 [50]. The official Landsat website indicates that pixels with NDSI < 0.7 should be classified as cloud [51]. In view of this range of threshold values, in this work, for the comparison of the MODIS cloud mask versus Landsat 7 and Landsat 8, two thresholds were selected: 0.4 and 0.7. Hence, a day is classified as Cloud if NDSI < 0.7 (or 0.4) and Clear if NDSI ≥ 0.7 (or 0.4).

We also carried out a visual inspection of a representative set of Landsat 8 and MODIS reflectance data. RGB composites were built using a red band and two Shortwave Infrared (SWIR) bands; these composites have been shown to render cloud pixels as white and snow pixels as red [29]. Landsat 8 RGB composites were built using the reflectance bands: R = band 4 (655 nm), G = band 6 (1609 nm), and B = band 7 (2201 nm). In the case of MODIS, we used MOD09GA reflectance data with R= band 1 (645 nm), G = band 6 (1640 nm), and B = band 7 (2130 nm).

#### 2.4.2. Albedo Filtering

A direct comparison of in-situ with satellite albedo is a difficult task, due to the different spatial and time resolutions of both measurements. Our in-situ albedo represented a circular area with a diameter of 30 m, whereas the MODIS albedo represented a square with a side of 500 m. The snow albedo at JG was quite homogeneous, as demonstrated by distributed albedo measurements carried out at Johnsons Glacier [52], so we assumed that in-situ measurements were representative of albedo over the area of a MODIS pixel. Furthermore, snow albedo is very sensitive to snow grain size, snowpack density, and water content, as well as to the amounts of direct and diffuse radiation. All of these factors can change in a few hours and, while AWS can provide several albedo measurements within a single day, only one albedo value per day at most can be obtained from a satellite sensor. Moreover, if we were to compare MOD10A1 with the in-situ data directly, taking into account the dates with both in-situ and MOD10A1 data only, the number of data points would be reduced dramatically. Because of this, in order to compare in-situ and MOD10A1 albedo using all the data available from the two datasets, we compared the albedo seasonality and the albedo trend obtained from the two datasets. We assumed that abrupt sudden changes in albedo were not feasible. The study area is permanently covered by snow or ice throughout the year, with no dirt from pollution and without patches of bare soil, such that albedo changes over time due to changing illumination conditions and/or to snow metamorphism, and neither of these can induce abrupt snow albedo changes. Data must be filtered in order to minimize the noise. In the case of in-situ albedo, we applied a 5-day windowed moving average. MODIS daily snow albedo data, however, exhibited much greater scattering than in-situ data, with some extraordinary and unrealistic low values. The scattering in the data could not be reduced sufficiently by applying a mean filter or a moving average, because the low values affect the filtered results. Taking into account that the MOD10A1 maximum values follow the same trend as in-situ data, MOD10A1 daily snow albedo was filtered using a maximum filter, such that the datum at a day *t_n_* was calculated as:
(4)$$\alpha_{MODIS}\left(t_{n}\right)=\max\left(\alpha_{MODIS}\left(t_{n-1}\right),\alpha_{MODIS}\left(t_{n}\right),\;\alpha_{MODIS}\left(t_{n+1}\right)\right),$$
where *t_n−1_*, *t_n_*, and *t_n+1_* are three consecutive dates of MODIS data (not necessarily three consecutive calendar dates).

#### 2.4.3. Albedo Seasonality

Regarding the time evolution of albedo through the melting season, we assumed that albedo decays according to an exponential law:(5)$$\alpha \left(t\right)=\alpha_{min}+\left(\alpha \left(0\right)-\alpha_{min}\right)e^{-\beta t},$$

Equation (5) is an adaptation of the snow albedo parameterization proposed in the Canadian Land Surface Scheme (CLASS) [53]. The use of this parameterization was based on previous theoretical and experimental studies. The snow albedo theoretically depends on grain size, as well as snow density [54]. The rate of growth of snow grains is a complicated function of water vapor movement, the initial geometry of snow crystals, and freeze-thaw cycles. Equation (5) was proposed by assuming that the magnitude of the snow albedo decreases exponentially with time, using an expression similar to the density decay [53], based on albedo data from several authors [55,56,57]. This parameterization has been used by other authors [58,59,60,61]. Furthermore, a comparison of prognostic models like Equation (5) against temperature-dependent models of snow albedo showed that the prognostic models were superior to the temperature-dependent ones [58].

In its original form, *α* in Equation (5) is assumed to decrease from a fresh snow value of 0.85, the value of the decay rate is fixed at *β* = 0.01 h^−1^ (0.24 day^−1^), and *α_min_* is set to 0.70 if no melting occurs and 0.50 if melting occurs. Other authors have used the same albedo decay with *β* = 0.03 day^−1^ and *α_min_* = 0.75 for snow metamorphism under dry conditions and *β* = 0.25 day^−1^ and *α_min_* = 0.50 for wet conditions [60]. An alternative approach consists of using the albedo decay of Equation (5) with *α*(0) = 0.85 and calibrating the values of α_min_ and β from experimental data [59]. In our case, for each season, we chose a time period along which a steady decrease of albedo was observed. Furthermore, taking into account that:(6)$$\underset{t\to \infty }{\lim}\alpha (t)=\alpha_{min},$$
the value of *α**_min_* was set equal to the minimum value observed along the period chosen, as long as this period was long enough for the albedo to attain a constant value, unless it suddenly increased due to a snowfall event. Then, both in-situ albedo and MOD10A1 albedo were fitted to Equation (5) and the values of the decay factor *β* and *α*(0) were obtained.

#### 2.4.4. Albedo Trend

The albedo trend was obtained by the robust statistical technique LOWESS (Locally Weighted Scatterplot Smoothing) [62,63], a non-parametric regression which minimizes the outliers of the dependent variable (albedo) in relation to the explanatory variable (time) based on neighboring points. For a given datum, the smoothed value is the value of a polynomial fit of the dataset, such that points close to that given datum are given more weight and the points that are further away from it are given less weight. The procedure is repeated on each datum to obtain all the smoothed values. In this work, LOWESS was applied on the filtered in-situ and MOD10A1 data, and a value of 2/3 was assigned for the smoother span, which gives the number of points which influence the smoothness at each value. This value was selected to avoid data that were very far apart in time from influencing one another and, in addition, to reduce the effect that outliers could cause. Moreover, the trend has to be evaluated such that the same time period is analyzed every season. Given the range of dates with albedo data (Table 2), the trend was evaluated taking into account albedo data from December 1 to April 10 each season (until February 11 2015 for the season 2014–2015), as data from September to November were missing in some seasons. The time was given in days, taking t = 0 as the first date with albedo data (December 1 2006). After the application of LOWESS, we obtained a set of smoothed albedo values. These values were then fitted to a straight line, the slope being the albedo increment per day.

## 3. Results and Discussion

### 3.1. Cloud Mask Performance

Out of the total number of days between December 1 2006 and February 11 2015 (2994) there were 1546 with MOD10A1 data classified as Snow, Land, or Cloud, 557 days with in-situ clr data, and 464 days with both in-situ clr and MOD10A1 data. The contingency table for MOD10A1 versus in-situ data for the cloud index clr = 0.7 are given in Table 4.

As we are interested in testing the capability of the cloud mask to distinguish snow and cloud, we focused our attention on the days which had a different classification in the two data sources. A day was classified as Clear by MOD10A1 but as Cloud by the in-situ clr on 15.7% of the total days analyzed. Conversely, a day was classified as Cloud by MOD10A1 but as Clear by in-situ data on 11.4% of the days.

The percentage of misclassified days given above is similar to that obtained in a similar study over Greenland [28], where the MOD10A1 cloud mask was assessed using in-situ data from five ground meteorological stations: Days classified as Clear by MOD10A1 were classified as cloudy by in-situ data on 11%, 5%, 5%, 4%, and 3% of the total days analyzed at the five stations. By contrast, when the stations reported clear sky, MOD10A1 reported cloud on 11%, 11%, 8%, 7%, and 4% of the days analyzed at the five stations. We conclude that the performance of MOD10A1 cloud mask over Livingston Island is similar to that over Greenland, where the MOD10A1 product has been used intensively over the past few years.

Contingency tables for the MOD10A1 cloud mask versus Landsat 7 and Landsat 8 NDSI are shown in Table 5 for two NDSI thresholds: 0.4 and 0.7. In the case of MOD10A1 versus Landsat 7, wrongly classified days amounted to 40.9% and 36.4% of the total of observations for NDSI = 0.4 and NDSI = 0.7, respectively. When taking Landsat 8 NDSI as ground truth, wrongly classified days were 32.2% and 25.8% of the total of observations for NDSI = 0.4 and NDSI = 0.7, respectively.

Let us focus our attention on the days that were given a different classification by Landsat 7 and Landsat 8 NDSI compared to the MOD10A1 cloud mask. In the case of Landsat 7, out of the 22 days with both Landsat 7 and in-situ data, 1 (4) were classified as Clear by MODIS and as Cloud by Landsat 7 using a threshold of 0.4 (0.7), and 1 (1) was classified by MODIS as Land. Out of the 62 dates with both MOD10A1 and Landsat 8 data, 7 (11) were classified as Clear by MODIS and as Cloud by Landsat 8 using a threshold of 0.4 (0.7). However, 6 (11) of these days were classified by MODIS as Land (not Snow). Furthermore, only 2 (0) of the days classified by MODIS as Land corresponded to Landsat 8 clear days for a threshold of 0.4 (0.7). This means that the agreement between MODIS and Landsat 8 was much better than it may seem from the results shown in Table 4, because the days classified as clear by MODIS included days classified as Snow and as Land. Let us remember that the study area was permanently covered by snow or ice, so days classified as Land could have been dates with a particular cloud cover. This is clearly seen in Figure 2, which shows RGB composites for MOD09GA and Landsat 8 data on March 28 2014. This date was classified as Cloud by Landsat 8 using both NDSI thresholds and Clear (Land, not Snow) by MODIS. The overall pattern in the data was very similar: While the most western and central parts of Livingston Island were cloud covered, the eastern part of the island was cloud free. The Landsat 8 image indicates the presence of discontinuous thick clouds. This date was classified as Land by MODIS.

December 15 2013 is an example of a day classified as Cloud by MODIS and Clear by Landsat 8, using both threshold values for NDSI (Figure 3). The Landsat 8 composites indicate the presence of thin clouds or fog (pink pixels) in the vicinity of both JCI and JG. In this case, the MODIS cloud mask was more conservative than the Landsat 8 NDSI. An example of a day classified Clear by MODIS and Landsat 8 is January 16 2014 (Figure 4). We conclude that both products agreed when the sky was completely cloud-free or completely overcast (not shown). Days with a thick and discontinuous cloud cover were classified as Cloud by Landsat 8 and as Land by MODIS. Days with thin clouds were classified as Clear by Landsat 8 and as Cloud by MODIS.

### 3.2. In-Situ Diurnal Albedo

As stated in Section 2.2, albedo increases with the SZA. The effect of the SZA is minimized by considering albedo values around local solar noon (Figure 5). The effect of the SZA is more noticeable on clear days. Moreover, the albedo on cloudy days is generally slightly higher than on clear days. The time span with no albedo data in Figure 5 corresponds to SZA > 90°. Taking into consideration albedo data around local noon, we ensure that albedo changes are due to snow metamorphization and/or down-welling radiation distribution, and not to the SZA.

### 3.3. Albedo Seasonality

In order to understand how the data were processed, we analyzed the general pattern of the original data (before filtering) from September 1 2007 to April 10 2009 (Figure 6). The other seasons followed a similar pattern.

Regarding in-situ data, we observed that:The scattering of the data was greater in September and April, and was minimal in summer months; probably due to a SZA effect.From September 1 to April 10, albedo followed a generally decreasing trend.Regarding MOD10A1 data, we observed that:MOD10A1 exhibited greater variability than in-situ data; a result similar to that obtained in Greenland [28], where it was found that MOD10A1 tracks the seasonal variability in the albedo but presents a greater variability than that observed in the terrestrial stations: Compared to a standard deviation of 0.033, 0.012, 0.012, 0.112, and 0.069, for the 16-day averaged albedo of the five AWS, the 16-day averaged MOD10A1 presented standard deviation values of 0.066, 0.042, 0.023, 0.097, and 0.083, respectively. This behavior agrees with that which we have obtained at JG.Some extremely low values were obtained.The maximum values (triangles in Figure 6b) followed the same trend as the in-situ data from September 1 to April 10. The triangles in Figure 6b represent the MOD10A1 maximum albedo every three consecutive values; that is to say, the snow albedo at *t_n_* (*α*(*t_n_*)) is represented by a triangle if it is bigger than *α*(*t_n-1_*) and *α*(*t_n+1_*). Triangles are joined by a solid black line, which provides a guide for the eye of the evolution of MOD10A1 maxima.

In order to minimize the spread of the data, the data were filtered, as explained in Section 2.4.2.

Boxplots (Figure 7) and histograms (Figure 8) were used to determine the data distribution and the effect of the filters. The mean, median, maximum, minimum, standard deviation (*σ*), and top and bottom values of the whiskers of the boxplots are given in Table 6. The 5-day window moving average applied on in-situ data had the effect of diminishing the spread of the data (*σ* is diminished, and the mean and the median were maintained). The original MOD10A1 data showed a great spread, with standard deviation σ = 0.15 and with two outliers (blue circles in Figure 7) corresponding to 11/28/2008, when a value of 0.30 was obtained (previous and next values were 0.78 and 0.86, respectively), and 03/11/2013, with a value of 0.24 (previous and next values were 0.96 and 0.85, respectively). The lower whisker of the original MOD10A1 albedo is longer than the upper one, an indication of the existence of a few extremely low values. No upper outliers were obtained, and the upper whisker top value coincided with the maximum albedo. This behavior can also be seen in the histograms. In the original MOD10A1 histogram, a lot of days with albedo values exceeding 0.90 were obtained, while days with albedo above 0.90 were very rare among the in-situ data. On the other hand, the minimum value of the original in-situ data was 0.62, while there were many days with an original MOD10A1 albedo below 0.6. From the data distribution, it is clear that the original MOD10A1 albedo exhibited a positive bias with respect to the original in-situ albedo, but also exhibited some extremely low values. The over-estimation of snow albedo from the MODIS data has been noted by other authors comparing the MCD43 albedo product against in-situ data [64] or against higher-resolution satellite data [65]. The maximum filter applied on the MOD10A1 eliminated the outliers and diminished σ to 0.11. It also increased the mean and the median. Thus, we expect the maximum filter to maintain the trend observed in the maximum values while, at the same time, eliminating the influence of unrealistic low values.

Seasonality of the albedo was studied using the filtered data. Equation (5) was linearized and the albedo decay was fitted for a given period of time. The value of α_min_ was set equal to the minimum value of the albedo during the time span considered in each case. The values of β (in day^−1^) and the intercept (ln(α(0)–α_min_)) are shown in Table 7, along with the time period during which a steady decay of the albedo was observed and the corresponding time span in days. This was the time span used for the fit. When fitting the albedo to Equation (5), we used t = 0 for the first date of the time interval along which albedo decay was observed and increased t from then on. All values were statistically significant with a level of significance of 5% (p-value < 0.05), except for those marked with an asterisk; the p-values of which are given at the bottom of Table 7. In Table 7, “No albedo decay” does not mean that no albedo decay occurred in a season, but that no albedo decay was observed in the time period over which data were available. It is worth noting that albedo decay takes place mostly from early September until January or February.

The *p*-value of *β* (MODIS) in the 2009–2010 season can be attributed to the fact that there was no decay, so we take *β* = 0 in this case. It is worth noting that the evolution of *β* followed the same trend for MOD10A1 and for in-situ data: There seems to be a deceleration in albedo decay from the season 2006–2007 onwards. Using the values of the intercept above, it is possible to calculate *α*(0) (Table 8). The rapid decrease of snow in the 2006–2007 season, as well as the low value of α_min_ in that season, agrees with the fact that, during that season, some ice surfaced at the location of the AWS [16].

The original, the filtered, and the fitted albedo for the seasons 2007–2008, 2008–2009, and 2013–2014 are shown in Figure 9. The vertical red lines in the figures on the left column of Figure 9 indicate the start and end dates of the albedo decay. The seasons 2007–2008 and 2008–2009 were selected because a clear albedo decay during a long time period was observed in both datasets. The season 2013–2014 was selected to show how the albedo decay could be observed in one data set, and not in the other. Similar figures for the rest of the seasons are provided in the Appendix A. We can see that the in-situ and MODIS data follow a very similar trend after filtering. In the 2006–2007 season, both in-situ and MODIS data decay from 1/12/2007 to 2/12/2007, increasing from then on. In the 2007–2008 season, both in-situ and MODIS data remain constant until October 24 2007, decrease steadily from October 24 2007 to January 16 2008, and remain constant (with a slight increase) again from January 16 2008 onwards. In the case of the 2008–2009 season, the behavior is similar, but the dates for the onset and end of the steady decay are different: Albedo seems to have decayed steadily from September 1 2008 to January 21 2009, remaining constant in the case of in-situ data and increasing in the case of MODIS after January 21 2009. In the 2009–2010 season, the MOD10A1 data exhibits a decreasing trend, starting on 9/01/2009. However, during the time period with in-situ data, the MOD10A1 data do not exhibit any clear trend, whereas in-situ data decrease very slightly from 12/08/2009 to 3/15/2010. In the 2010–2011 season, in-situ data are available from 1/02/2011, and both in-situ and MOD10A1 data exhibit an increasing trend from that date onward. However, MOD10A1 data display a clear decreasing trend from 9/01/2010 to 1/10/2011. In the 2013–2014 season we observe a clear decay in MODIS filtered data from September 7 2013 to January 30 2014. In this season, no in-situ data were available during the albedo decay period. In the 2014–2015 season, in-situ data were available from 12/22/2014 and exhibit a clear decay from that date to 2/11/2015. The right column in Figure 9 shows the fitting of filtered in-situ and MOD10A1 data to Equation (5), with time given in days and t = 0 at the onset of the albedo decay. The data are fitted from the onset of the decay until the end of it. In-situ data were not fitted to Equation (5) in the seasons 2010–2011 and 2013–2014, as no albedo decay was observed during the time the data were available.

Our results show an early onset of decay (early September) and a duration between 85 days for the 2007–2008 season and 167 for the 2013–2014 season (we did not take into account the decay duration of the 2006–2007 season, because we have no in-situ data before December 1 2006). Albedo decay was not observed when data were not available early in the season. The decay factor (β) obtained at Johnsons Glacier was in the range of 0–0.049 day^−1^ for in-situ albedo and 0–0.094 day^−1^ for MOD10A1 albedo. The seasonal behavior of snow albedo can be explained by taking into account snowmelt from September to April. The onset of melting has been noted, on average, to be in early to mid-October in the Antarctic Peninsula and the SSI in the period 2000–2009 [15]. The average duration of the melt season in the SSI in the period 2000–2009 has been noted as 125 days. The evolution of albedo over the melting season has been parameterized using the exponential decay of Equation (5) for several locations in Colorado (USA) and Rhône-Alpes (France) [59]. The decay factors obtained were 0.003 h^−1^ (0.072 day^−1^), 0.004 h^−1^ (0.096 day^−1^), and 0.005 h^−1^ (0.12 day^−1^) in Colorado and 0.005 h^−1^ (0.12 day^−1^) at Rhône-Alpes, with an all-site average of 0.004 h^−1^ (0.096 day^−1^). These results are as expected, since albedo decay in Antarctic areas is expected to be much weaker than in other areas of the Earth. We calculated the Root Mean Square Error (RMSE) of the fit to Equation (5) using the equation:(7)$$RMSE=\sqrt{\frac{\sum_{i=1}^{N}{\left(\alpha_{i,f}-\alpha_{i,o}\right)}^{2}}{N}},$$
where *α**_i,f_* and *α**_i,0_* are the fitted and observed albedo, respectively, and N is the number of albedo data over the time span considered. The RMSE values, as shown in Table 8, vary from 0.02 to 0.03 for the in-situ albedo and from 0.015 to 0.08 for the MOD10A1 albedo. The RMSE obtained by Malik et al. [59] were 0.096, 0.090, 0.076, and 0.068 for the locations in Colorado and 0.038 at Rhône-Alpes. Other authors have used similar parameterizations for albedo over Greenland, obtaining a much higher RMSE [61]; from 0.10 to 0.23. Our results show that the albedo decay at Johnsons Glacier can be described by the exponential law proposed in Equation (5).

### 3.4. Albedo Trend

The trend of the in-situ moving average and the MOD10A1 maxima for the time period from December 1 2006 to February 11 2015 along was calculated (Figure 10). The shaded areas in Figure 10 correspond to dates from April 10 to September 1 in each year.

Regarding the trend, both the MOD10A1 and in-situ data exhibited a small increase in the period studied, which was more pronounced in the MODIS data, although these also showed greater fluctuation. Table 9 shows the regression values of the albedo trend.

In-situ data exhibited an increase of 0.00000606 per day, while MOD10A1 data exhibited an increase of 0.0000195 per day. These daily increases amounted to an increase of 0.02 (2%) in in-situ albedo and 0.06 (6%) in MOD10A1 albedo from December 2006 to February 2015. The discrepancy between the snow albedo increase from in-situ and MOD10A1 data could be due to spatial heterogeneity, but this point needs further research.

The increase of snow albedo could be due to a reduction of summer melting. It is well-known that snowmelt decreases albedo [24], due to the lower reflectance of liquid water. Thus, a decrease of summer melting must induce an increase of snow albedo. Recent investigations have shown that the average thickness of the snow layer has increased in the surroundings of JCI in the period 2008–2016 [9]. The number of days with snow at JCI has also increased in the period 2008–2016 [9]. The net yearly mass balance of Jonhsons Glacier was negative from 2002 to 2007 and positive from 2008 to 2016 [9]. These findings have been attributed to an increase of snow accumulation in winter accompanied by a lower summer melting. According to simulations of the surface mass balance over the Antarctic Peninsula and the South Shetland Islands, in the period 1979–2016, the snowmelt decreased over the period 1979–2016 [8]. This decrease has been more acute in the last decade, as a result of widespread cooling over most of the Antarctic Peninsula [8,66]. The snow cover evolution in the Limnopolar CALM site over the period 2009–2014 has also been investigated [10]. This CALM site is located in the Byers peninsula, on Livingston Island, at about 37 km (in a straight line) from JCI Station. An increase in snow cover duration, along with a reduction in the melting season was observed. All these results agree with the increasing trend in albedo which we have observed in both in-situ and MOD10A1 data.

## 4. Conclusions

The seasonality and the trend of snow albedo on Livingston Island, Antarctica, have been analyzed using in-situ and MODIS data. We have shown that both datasets exhibit the same trend and seasonality when properly filtered. The in-situ data were filtered using a 5-day windowed moving average while the MOD10A1 data were filtered using a maximum filter. This opens the possibility for using the MODIS daily snow albedo product MOD10A1 to characterize snow albedo seasonality and trend over Antarctica, where this product has never been tested before. Snow albedo seasonal behavior was analyzed during the melting season (from September 1 to April 10) between the 2006–2007 season to the 2014–2015 season. Snow albedo decays exponentially. The onset and duration of the decay varied from season to season, with a maximum decay rate in the 2006–2007 season. The decay duration varied between 167 days in the 2013–2014 season and 85 days in the 2007–2008 season. The albedo trend was also analyzed using the filtered data. A slight, but statistically significant, increase of albedo was obtained from 2006 to 2015; a total increase of 0.02 (2%) in in-situ albedo and 0.06 (6%) in MOD10A1 albedo. In order to complete our study, and as the MODIS MOD10A1 daily snow albedo product has not previously been assessed in Antarctica before, we assessed the cloud mask performance. The cloud mask output of the MODIS MOD10A1 product was compared to the classification obtained using an in-situ cloud index and to that using the NDSI from Landsat 7 and Landsat 8 imagery. The results show that the MOD10A1 cloud mask output agrees with the data obtained from in-situ data and through Landsat 7 and Landsat 8 NDSI classification. Moreover, the performance is comparable to that obtained over Greenland.

We conclude that MOD10A1 snow albedo shows a temporal behavior, similar to that of in-situ albedo. We conclude that MOD10A1 daily albedo can be used to study the time evolution of albedo on Livingston Island. Further research is underway, aiming to study the snow albedo trend and seasonality over the entirety of Livingston Island and the Antarctic Peninsula and its effects on the mass and energy balances of the glaciers of the island. We have also recently obtained albedo distributed measurements in the area, aiming to study the impact of snow albedo spatial heterogeneity. Further research is being conducted to understand why MOD10A1 albedo, when filtered with a maximum filter, reproduces the behavior of in-situ albedo.

## Figures and Tables

**Figure 1 sensors-19-03569-f001:**
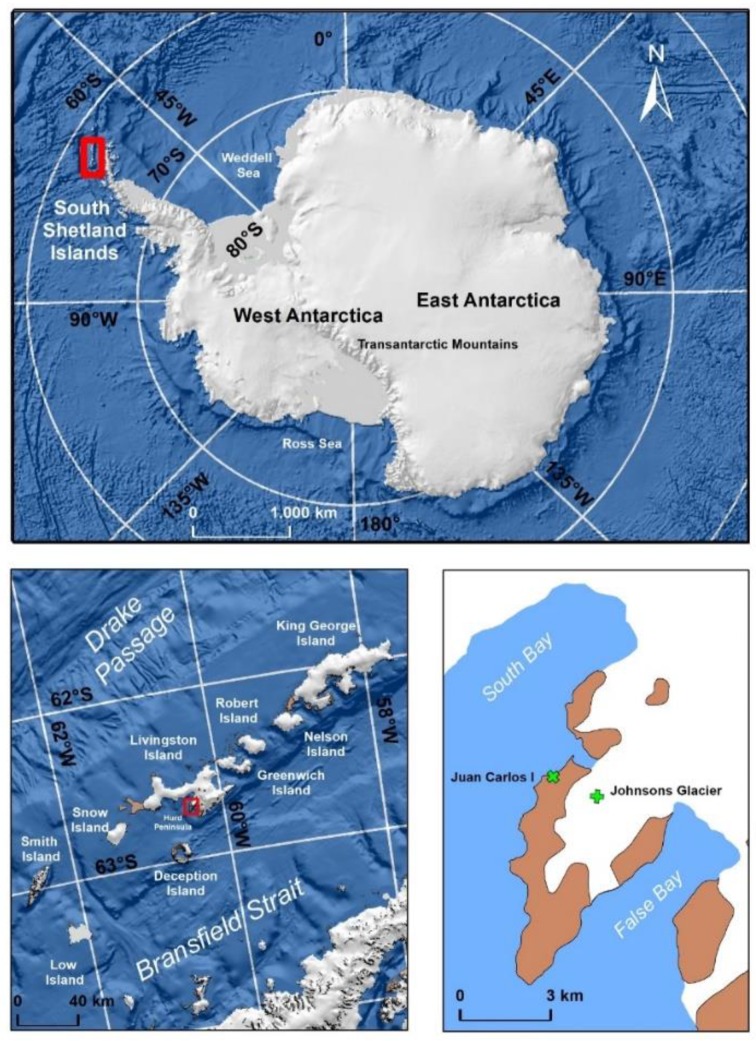
Location of the South Shetland Islands (red square, top image), and the study area on Livingston Island (red square, left hand bottom image). The locations of the Automatic Weather Stations are shown in the right bottom image.

**Figure 2 sensors-19-03569-f002:**
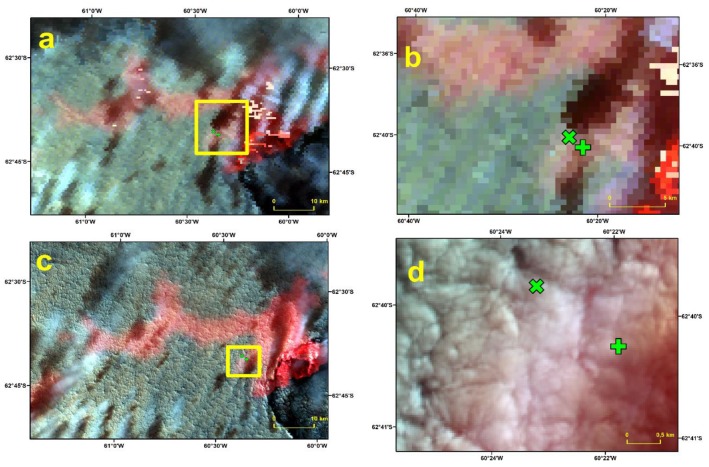
RGB composites of MOD09GA (**a**) and Landsat 8 (**c**) of Livingston Island of March 28 2014. The yellow box shows the area of the close view of the RGB composite of MOD09GA (**b**) and Landsat 8 (**d**) in the vicinity of the Automatic Weather Stations. The location of the Automatic Weather Stations at Juan Carlos I and Johnsons Glacier are indicated with a green cross and a green plus sign, respectively. Images provided in the WGS84 UTM20S projection.

**Figure 3 sensors-19-03569-f003:**
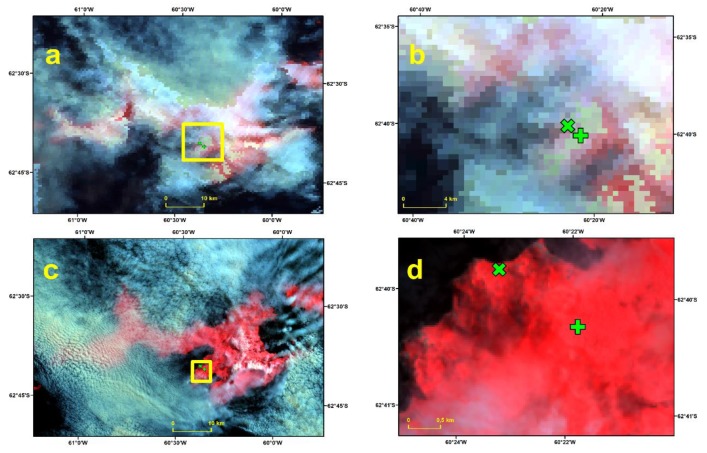
RGB composites of MOD09GA (**a**) and Landsat 8 (**c**) of Livingston Island of December 15 2013. The yellow box shows the area of the close view of the RGB composite of MOD09GA (**b**) and Landsat 8 (**d**) from December 15 2013 in the vicinity of the Automatic Weather Stations. The location of the Automatic Weather Stations at Juan Carlos I and Johnsons Glacier are indicated with a green cross and a green plus sign, respectively. Images provided in the WGS84 UTM20S projection.

**Figure 4 sensors-19-03569-f004:**
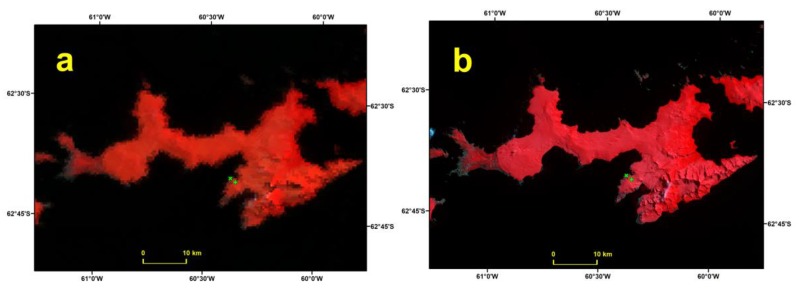
RGB composites of MOD09GA (**a**) and Landsat 8 (**b**) of Livingston Island of January 16 2014. Images are projected in WGS-84 UTM20S. The location of the Automatic Weather Stations at Juan Carlos I and Johnsons Glacier are indicated with a green cross and a green plus sign, respectively.

**Figure 5 sensors-19-03569-f005:**
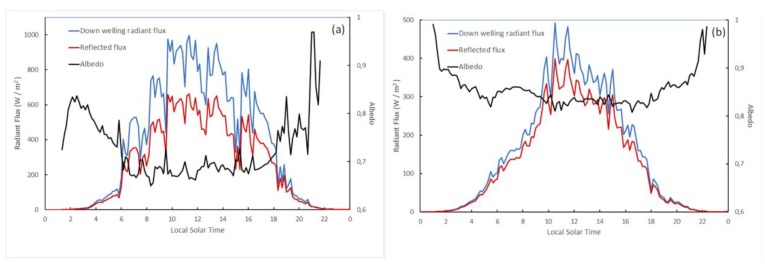
Diurnal evolution of the down welling radiant flux (solid blue line), the reflected radiant flux (solid red line), and the albedo (solid black line) at Johnsons Glacier. Day with clear sky (December 24 2009) (**a**) and an overcast day (December 22 2011) (**b**). Albedo and radiation values are given every 10 min.

**Figure 6 sensors-19-03569-f006:**
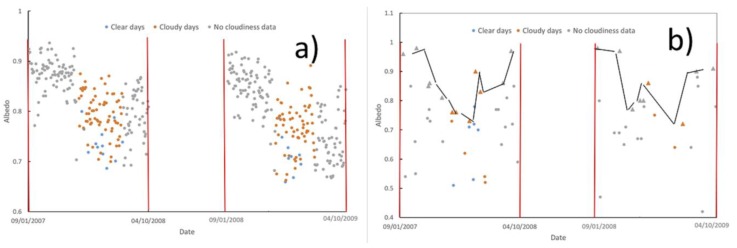
In-situ (**a**) and MOD10A1 (**b**) albedo at JG from September 1 2007 to April 10 2009. In-situ data are calculated as the mean value from 10 h to 14 h LST. Days are classified according to the in-situ cloudiness index, as Clear (blue symbols) or Cloudy (orange symbols). Days with no cloudiness data are represented in grey symbols. Triangles in (**b**) represent the maximum value for every three consecutive data. The trend of MOD10A1 maxima is indicated with a solid black line in (**b**). Red vertical lines indicate, from left to right: 09/01/2007, 04/10/2008, 09/01/2008, and 04/10/2009.

**Figure 7 sensors-19-03569-f007:**
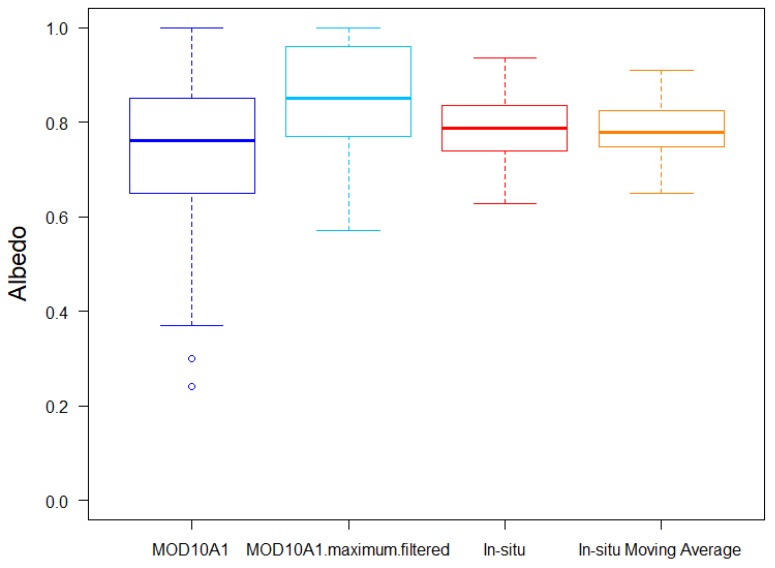
Boxplot of the original MOD10A1 albedo data (dark blue box), MOD10A1 albedo data filtered with the maximum filter (light blue box), original in-situ albedo (red box), and in-situ albedo filtered with a 5-day window moving average (orange box). Circles represent outliers.

**Figure 8 sensors-19-03569-f008:**
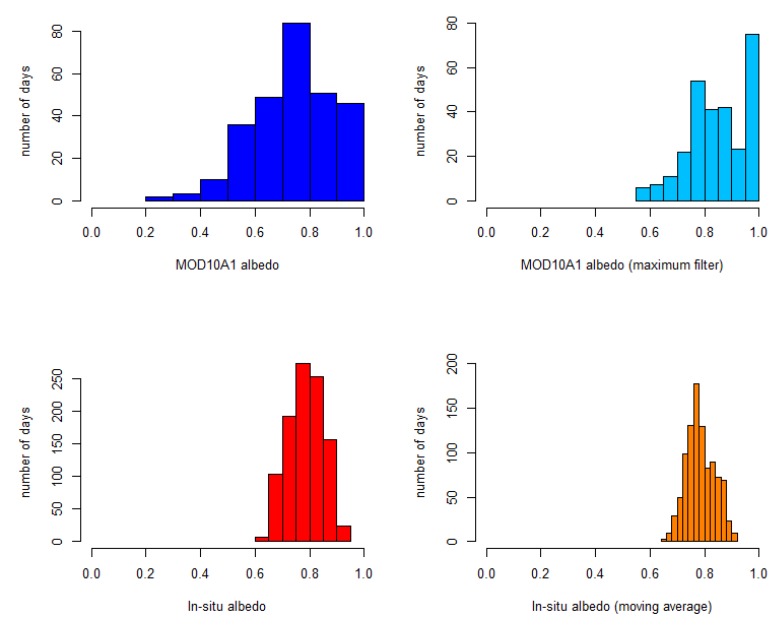
Histograms of in-situ and MOD10A1 albedo before and after filtering: MOD10A1 albedo (top left), MOD10A1 albedo (maximum filter) (top right), in-situ albedo (bottom left) and in-situ albedo (moving average) (bottom right).

**Figure 9 sensors-19-03569-f009:**
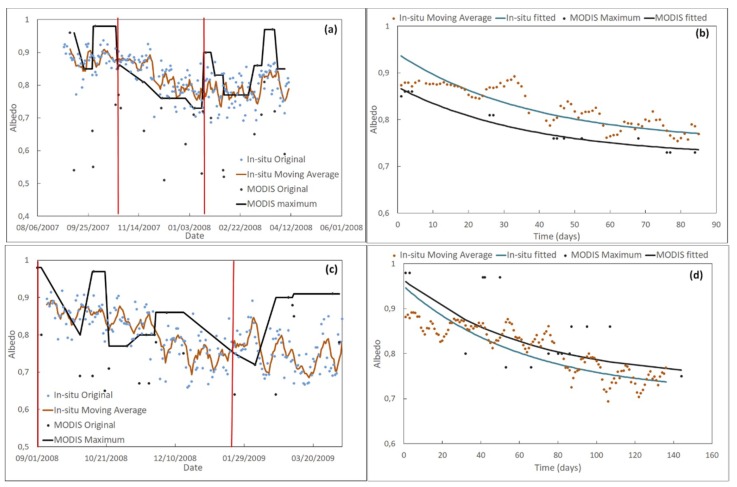
Left column: original in-situ albedo (blue dots), original MOD10A1 albedo (black dots), in-situ albedo moving average (brown solid line) and filtered MOD10A1 albedo (black solid line) from September 1 to April 10. Right column: in-situ albedo moving average (brown dots) and filtered MOD10A1 albedo and the corresponding fit to the exponential decay. Season 2007–2008 (**a**,**b**), season 2008–2009 (**c**,**d**), season 2013–2014 (**e**,**f**). The vertical red lines in (**a**,**c**,**e**) indicate the start and end dates of the snow melting.

**Figure 10 sensors-19-03569-f010:**
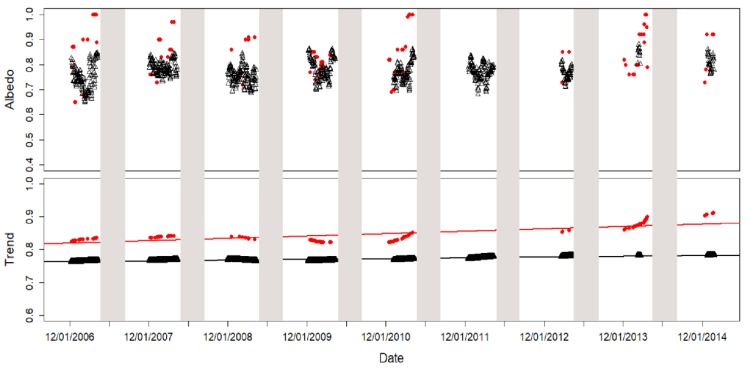
Moving average of in-situ albedo (open triangles) and maximum values of MOD10A1 albedo (red dots) from December 1 to April 10 (upper figure) and the calculated trends of in-situ moving average (open triangles) and MOD10A1 maximum albedo (red dots) from December 1 to April 10 (bottom figure). The linear fits of the trend are also shown. The shaded areas correspond to dates from April 10 to September 1 each year.

**Table 1 sensors-19-03569-t001:** Number of days with MOD10A1 (V005), Landsat, and in-situ data from December 1 2006 to February 11 2015. MOD10A1 data corresponds to the pixel where JG is located. For each season, only dates from September 1 until April 10 are considered (December 1 to April 10 for the 2006–2007 season and from September 1 to February 11 in 2014–2015 season). JCI, automatic weather station (AWS) at Juan Carlos I station (in-situ diffuse and global irradiances); JG = AWS at Johnsons Glacier (in-situ albedo). Shaded cells indicate data that have not been crossed.

-	MOD10A1 (Cloud, Land, Snow)	MOD10A1 (Snow Albedo)	JCI (In-Situ Irradiance)	JG (In-Situ Albedo)	L7	L8
MOD10A1 (Cloud, Land, Snow)	1546	286	464	-	22	62
MOD10A1 (Snow Albedo)	286	286	-	159	-	-
JCI (In-Situ Irradiance)	464	-	557	-	-	-
JG (In-Situ Albedo)	-	159	-	1008	-	-

**Table 2 sensors-19-03569-t002:** Range of dates with MOD10A1 and in-situ albedo at Johnsons Glacier.

Season	MOD10A1	In-Situ
2006–2007	9/1/2006–4/10/2007	12/1/2006–4/10/2007
2007–2008	9/1/2007–4/10/2008	9/1/2007–4/10/2008
2008–2009	9/1/2008–4/10/2009	9/1/2008–4/10/2009
2009–2010	9/1/2009–4/10/2010	12/1/2009–4/10/2010
2010–2011	9/1/2010–4/10/2011	1/1/2011–4/10/2008
2011–2012	No Data	12/14/2011–4/10/2012
2012–2013	1/1/2013–4/10/2014	2/14/2013–4/10/2013
2013–2014	2/1/2014–4/10/2014	2/1/2014–3/1/2014
2014–2015	11/1/2014–4/10/2015	12/22/2014–2/11/2015

**Table 3 sensors-19-03569-t003:** Range of dates with in-situ diffuse and global radiation from the AWS at JCI.

**Season**	**2006–2007**	**2007–2008**	**2008–2009**	**2009–2010**	**2010–2011**
Dates Range	1/12/2006–6/03/2007	4/12/2007–22/2/2008	1/12/2008–12/2/2009	14/12/2009–7/03/2010	3/01/2011–24/02/2011
**Season**	**2011–2012**	**2012–2013**	**2013–2014**	**2014–2015**	**-**
Dates Range	4/12/2011–23/02/2012	26/12/2012–20/01/2013	3/02/2014–19/02/2014	2/12/2014–25/01/2015	-

**Table 4 sensors-19-03569-t004:** Contingency table of MOD10A1 cloud mask versus in-situ cloud index for clr = 0.7.

	In Situ clr	Cloud	Clear	Total
MOD10A1				
Cloud		313	53	366
Clear		73	25	98
Total		386	78	464

**Table 5 sensors-19-03569-t005:** Contingency table for MOD10A1 cloud mask versus NDSI of Landsat 7 and Landsat 8 for NDSI thresholds 0.4 and 0.7. Cd, Number of days classified as Cloud; Cr, Number of days classified as Clear; T, Total number of days.

-	Landsat 7 NDSI Threshold						Landsat 8 NDSI Threshold					
-	0.4			0.7			0.4			0.7		
-	Cd	Cr	T	Cd	Cr	T	Cd	Cr	T	Cd	Cr	T
MOD10A1												
Cd	9	8	17	13	4	17	31	13	44	39	5	44
Cr	1	4	5	4	1	5	7	11	18	11	7	18
T	10	12	22	17	5	22	38	24	62	50	12	62

**Table 6 sensors-19-03569-t006:** Statistical data of in-situ and MOD10A1 albedo data before and after filtering. σ is the standard deviation.

-	In-Situ Original	In-Situ Filtered	MOD10A1 Original	MOD10A1 Filtered
Maximum	0.94	0.91	1.00	1.00
Minimum	0.63	0.65	0.24	0.57
Mean	0.79	0.79	0.75	0.86
Median	0.79	0.78	0.76	0.85
*σ*	0.06	0.05	0.15	0.11
Upper Whisker Maximum	0.94	0.91	1.00	1.00
Lower Whisker Minimum	0.63	0.65	0.37	0.57

**Table 7 sensors-19-03569-t007:** Albedo decay characterization for each melting season. Time periods of albedo decay, duration of the periods, decay constant *β*, and intercept (ln(*α*(0)–*α_min_*)) obtained from the linear fit for in-situ and MOD10A1 albedo data. Dates read as mm/dd/yyyy. For each season, decay constant *β* and intercept values for in-situ and MOD10A1 albedo data are shown in the upper and lower line, respectively.

Season	Decay Duration (days) Time Period	β (day^−^^1^) In-Situ MODIS	Intercept In-Situ MODIS
2006–2007	31	0.049 ± 0.009	−1.88 ± 0.16
	1/12/2007–2/12/2007	0.094 ± 0.019	−1.2 ± 0.4
2007–2008	85	0.026 ± 0.002	−1.66 ± 0.11
	10/24/2007–1/16/2008	0.026 ± 0.004	−1.9 ± 0.2
2008–2009	143	0.0159 ± 0.0012	−1.43 ± 0.09
	9/01/2008–1/21/2009	0.016 ± 0.005	−1.4 ± 0.4
2009–2010	98	0.011 ± 0.002	−2.29 ± 0.15
	12/08/2009–3/15/2010	0.002 ^1,^* ± 0.007	−2.2 ± 0.4
2010–2011	124	No Albedo Decay	No Albedo Decay
	9/08/2010–1/10/2011	0.014 ± 0.004	−0.8 ± 0.3
2011–2012	-	No Albedo Decay	No Albedo Decay
		No Data	No Data
2012–2013	-	No Albedo Decay	No Albedo Decay
		No Albedo Decay	No Albedo Decay
2013–2014	167	No Albedo Decay	No Albedo Decay
	9/07/2013–1/30/2014	0.017 ± 0.002	−1.29 ± 0.15
2014–2015	48	0.041 ± 0.007	−1.82 ± 0.19
	12/26/2014–2/11/2015	0.033 ± 0.002 **	−1.7 ± 0.8 ***

^1^*p*-value for the values with asterisks are: (*) 0.78, (**) 0.18, (***) 0.08.

**Table 8 sensors-19-03569-t008:** Values of the initial (*α*(0)) and minimum (*α_min_*) values of albedo during the albedo decay for each season, for in-situ and MOD10A1 albedo data. The Root Mean Square Error (RMSE) is also given.

Season	*α*(0) In-Situ/MODIS	*α**_min_* In-Situ/MODIS	RMSE In-Situ/MODIS
2006–2007	0.79/0.86	0.64/0.57	0.02/0.05
2007–2008	0.94/0.87	0.75/0.72	0.02/0.015
2008–2009	0.95/0.98	0.71/0.74	0.03/0.07
2009–2010	0.80/0.78	0.70/0.67	0.03/0.05
2010–2011	-/1.0	-/0.59	-/0.08
2013–2014	- /1.0	-/0.74	-/0.06
2014–2015	0.86/0.91	0.70/0.72	0.02/0.06

**Table 9 sensors-19-03569-t009:** Slope and intercept of the linear fit of the albedo trend values versus time.

-	In-Situ	MOD10A1	*p*-Value
Intercept	0.76433 ± 00016	0.821 ± 0.003	<2 × 10^−16^
Slope (day^−^^1^)	0.00000606 ± 0.00000012	0.0000195 ± 0.0000017	<2 × 10^−16^

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
