# Peer review of "Snow Albedo Seasonality and Trend from MODIS Sensor and Ground Data at Johnsons Glacier, Livingston Island, Maritime Antarctica"

_sensors, 2019, doi:10.3390/s19163569_

Round 1

Reviewer 1 Report

Comments to the authors of the paper: Relating satellite snow albedo trend and seasonality to ground measurements at Johnsons Glacier, Livingston Island, Maritime Antarctica

General comments:

This paper tries to present that the MODIS snow albedo daily product (MOD10A1) is able to capture snow albedo seasonality and trend under all sky conditions, in conjunction with using the in-situ measurements at Johnsons Glacier (Livingston Island, Antarctica). Generally, I feel part of conclusions in this manuscript is probably interesting for potential readers, e.g., a statistically significant slight increase of albedo was observed from 2006 to 2015, especially from the in-situ measurements. I feel that the manuscript is not easy to follow and I suggest authors consider a general flow chart for organizing this work. I 'd like suggest major revisions before accepting this paper for publication. My major comments are as follows:

1.    One of my major concerns comes from the use of the MODIS snow albedo daily product (MOD10A1). Current MODIS BRDF/Albedo product suite (MCD43A, Collection 6) is offering a daily product of the intrinsic albedos (i.e., BAS and WSA). What are possible reasons that this manuscript didn’t choose to use this MCD43A product suite? Some early work clearly pointed out that the MODIS BRDF/Albedo product probably has a better quality than MOD10A1 (e.g., Stroeve et al., 2006, RSE; Hall et al., https://ntrs.nasa.gov/search.jsp?R=20110007072). Moreover, this work didn’t use the latest version of MOD10A1 (i.e., V6), and explained that “because it has never been used in Antarctica before and the scientific literature is more abundant”. I don’t think this is a reasonably acceptable reason since V6 corrects a drift in snow albedo trend from V5, which happens to be within the aim that this paper works on, although authors of the manuscript argue that this correction was just based on the investigation over the Greenland Ice Sheet. After all, the increasing trend of albedo that this paper has explored by using both the MOD10A1 and in-situ measurements is quite weak, although such a trend is statistically significant in terms of this manuscript.

2.    Another major concern is from the probably large uncertainties using the maximum filter to process the MOD10A1. The average of one categorical dataset should be theoretically a reasonable representativeness for the trend of that dataset, but the maximum and minimum are most probably the outliers. Although this manuscript shows that the maximum of the MOD10A1 captures a possible trend similar with that using the in-situ measurements. This is not very convincible to me since there is not a reasonably acceptable explanation explored for the reason why maximum of the MOD10A1 is effective in the paper, although authors may argue that they will address this in future. However, I feel that this is very important for understanding the conclusion of this paper.

3.    It is not very clear to me about some descriptions on snow melting in this manuscript. For example, the duration of the decay of snow melting increases, i.e., “The duration of the decay varies from 85 days (2007-2008) to 167 days (2013-2014)” in Line 25; however, the snow melting decreases, i.e., “a decrease of snow melting in the study area” in Line 27. Is it possible that snow melting decreases with the duration of the decay of snow melting increasing? Most of Antarctica is covered by ice permanently, I guess, which includes Johnsons Glacier as well. Now that the duration of the decay of snow melting increases, what cause the snow melting decreasing? It seems to me that these two events are somewhat in conflict with each other. I think readers may need more details for understanding such statements.

Minor comments,

  1. Line 17-19 for “Because…cloud mask”: I don't understand the logics in this sentence? In other word, if this product was used in Antarctic before, you don't have to assess the performance of the MOD10A1 cloud mask, right?
  2. Line 68-69 for “It is well known that cloud cover normally increases the spectrally integrated albedo”. For what? I guess this is most probably for vegetation.
  3. Line 72-74 for “However…including overcast days”: Why do you propose such a question? What is the scientific meaning of this question? I am not surprised that it can be tracked even just using the nadir-view reflectance that can capture the major reflectance magnitude of albedo. To monitor the temporal changes, I don't think this necessarly needs a high absolute accruacy of albedo.
  4. Line 75-82: I feel that this segment is written in confusion. Need to rephrase it.
  5. Line 88-90: Which studies? Need to cite some references here.
  6. Line 152: I think this is probably not an acceptable reason why this paper didn't use V6.
  7. Line 285-286: It is not clear to me how these two values were used at the same time for one scene of Landsat imagery.
  8. Line 298-299: You cannot simply assume that in-situ measurements are representative of albedo over the area of a MODIS pixel without any assessment of the representativeness of the in-situ measurements.
  9. Line 314-316: I think these extreme values are most probably caused by the uncertainty of the algorithm. This is why I don’t suggest use the MOD10A1.
  10. Line 448-450: This is no evidences to present the albedo changes as a function of snow metamorphization. Is this guess really reasonable? Why it is not caused by sub-pixel clouds?
  11. Line 463-465: Please highlight the duration from September 1 to April 10 in your figure. I don't observe such an increasing trend.
  12. Line 480: “de” typo?
  13. Figure 8: Indeed, a possibly somewhat similar trend exists between these two figures, but these points are so scattering.

Reviewer 2 Report

Please see attached report

Round 2

Reviewer 1 Report

1. The response of the authors to my comments and suggestions and the revised version generally make sense to me. The use of MOD10A1 is a possible choice and is allowed for potential users for their more explorations. To support this paper for not using MCD43A snow albedo, it should be helpful for authors to cite the latest papers in this manuscript as evidences (Jiao et al., 2019, RSE; Ding et al., 2019, RS) as follows. These two papers has developed the RTLSR model that is not necessarily appropriate for modeling snow BRDFs, and moreover presented that the RTLSR model somewhat underestimates snow albedo to some degree. 

     Jiao, Z. et al. (2019). Development of a snow kernel to better model the anisotropic reflectance of pure snow in a kernel-driven BRDF model framework. Remote Sens. Environ. 221 (2019) 198–209.

     Ding, A., et al. (2019). Evaluation of the Snow Albedo Retrieved from the Snow Kernel Improved the Ross-Roujean BRDF Model. Remote Sens. 2019, 11,1611.

2. The response " A way to get rid of these outliers keeping the trend of the maximum values is to apply a maximum filter". If there is any paper to use a maximum filter for snow, such references are actually needed. In my memory, there are some papers using maximum filter such as for NDVIs, which is however due to different principle, e.g., for removing subpixel cloud etc.  
